# Neodymium-Facilitated Visualization of Extreme Phosphate Accumulation in Fibroblast Filopodia: Implications for Intercellular and Cell–Matrix Interactions

**DOI:** 10.3390/ijms252011076

**Published:** 2024-10-15

**Authors:** Marina Kravchik, Anastasia Subbot, Airat Bilyalov, Ivan Novikov, Ruslan Deviatiiarov, Yusef Yusef, Oleg Gusev

**Affiliations:** 1Federal State Budgetary Institution of Science “M.M. Krasnov Research Institute of Eye Diseases” (Krasnov Research Institute of Eye Diseases), 119021 Moscow, Russia; kletkagb@gmail.com (A.S.);; 2Institute of Fundamental Medicine and Biology, Kazan Federal University, 420008 Kazan, Russiao.gusev.fo@juntendo.ac.jp (O.G.); 3SBHI Moscow Clinical Scientific Center Named after Loginov MHD, 111123 Moscow, Russia; 4Institute of Biochemistry and Genetics, Ufa Federal Research Centre of Russian Academy of Sciences, 450054 Ufa, Russia; 5Life Improvement by Future Technologies (LIFT) Center, 121205 Moscow, Russia; 6Intractable Disease Research Center, Graduate School of Medicine, Juntendo University, Tokyo 113-8421, Japan

**Keywords:** lanthanoid staining, scanning electron microscopy, phosphate anion, cell–matrix interaction, intercellular interaction, filopodia, apoptosis, mitochondria, lanthanoids, neodymium

## Abstract

A comprehensive understanding of intercellular and cell–matrix interactions is essential for advancing our knowledge of cell biology. Existing techniques, such as fluorescence microscopy and electron microscopy, face limitations in resolution and sample preparation. Supravital lanthanoid staining provides new opportunities for detailed visualization of cellular metabolism and intercellular interactions. This study aims to describe the structure, elemental chemical, and probable origin of zones of extreme lanthanoid (neodymium) accumulation that form during preparation for scanning electron microscopy (SEM) analysis in corneal fibroblasts filopodia. The results identified three morphological patterns of neodymium staining in fibroblast filopodia, each exhibiting asymmetric staining within a thin, sharp, and extremely bright barrier zone, located perpendicular to the filopodia axis. Semi-quantitative chemical analyses showed neodymium-labeled non-linear phosphorus distribution within filopodia, potentially indicating varying phosphate anion concentrations and extreme phosphate accumulation at a physical or physicochemical barrier. Phosphorus zones labeled with neodymium did not correspond to mitochondrial clusters. During apoptosis, the number of filopodia with extreme and asymmetric phosphorus accumulation increases. Supravital lanthanoid staining coupled with SEM allows detailed visualization of intercellular and cell–matrix interactions with high contrast and resolution. These results enhance our understanding of phosphate anion accumulation and transfer mechanisms in cells under normal conditions and during apoptosis.

## 1. Introduction

Information regarding the interaction and exchange of substances between cells under various physiological and pathological conditions is crucial for understanding numerous aspects of cellular biology. In recent years, research into cell–matrix interactions via filopodia and long-distance intercellular communication through specialized cellular structures such as filopodia, tunneling nanotubes, and similar structures has received increased attention [1,2,3,4,5,6,7,8,9].

Various techniques are employed to investigate the interactions that take place through these structures. Fluorescence microscopy enables us to monitor exchange processes in real time, but its limitation lies in its low resolution, which prevents us from examining the ultrastructure in great detail [10,11]. In contrast, classical scanning and transmission electron microscopy allow for a detailed examination of cellular ultrastructure due to their high resolution, enabling the visualization of the smallest details in structures. However, the disadvantages of these methods include the complex and aggressive sample preparation process, which can alter the natural state of the samples. Cryo-electron microscopy, on the other hand, enables the visualization of three-dimensional structures of thin cellular components at nanoscale resolutions in frozen samples [12]. While this technique offers the benefit of obtaining high-resolution three-dimensional images, it also presents challenges in terms of complexity and the need for specialized sample preparation methods [13].

The method of lanthanoid staining coupled with scanning electron microscopy (SEM) is notable. This approach, based on the rapid visualization method described in our previous works [14,15,16,17], involves treating the cells with physiologically buffered solution containing ions of relatively heavy lanthanoids. This method does not require traditional SEM fixation and complete dehydration of the samples, simplifying the preparation process and preserving cellular structures. The staining by relatively heavy lanthanoids also provides high-contrast SEM images without the need for additional heavy metal sputtering, which is common in conventional SEM. A particularly important aspect of this method is that the relatively heavy lanthanoids bind to intracellular components, enabling their visualization by SEM using a back-scattered electron (BSE) detector, which allows for the observation of not only surface structures but also subsurface components.

The supravital staining of cells with lanthanoids such as neodymium induces a temporary ametabolic state. This state effectively halts intracellular processes, such as the movement of organelles and RNA transcription, without causing permanent damage to the cell. By capturing cells in this ametabolic state, the technique allows for the detailed observation of cellular metabolism and morphology at a specific point in time, effectively providing a “snapshot” of cellular activity. This process is described in Subbot et al. [17], who demonstrated that lanthanoid staining preserves the cell in a reversible ametabolic condition. This effect makes it possible to study the cell in a static metabolic state, revealing the kinetics of cellular processes by effectively pausing them at a given time.

While developing the lanthanoid staining technique, our team found some extremely bright and contrast locations within the cell that could not be easily identified as any known process or structure. One such structure (or pseudo structure) is associated with filopodia and filopodia-like formations that establish distant contacts in different types of cells, including fibroblasts, epithelial cells, and limbal stem cells. Our team observed that these areas contain zones of extreme accumulation of lanthanoid (neodymium) dye.

Neodymium was selected as the lanthanoid of choice after our previous extensive empirical testing of various lanthanoids. Among the tested lanthanoids, neodymium provided the maximum brightness and contrast in BSE imaging. We did not specifically investigate the underlying reasons for this, as our choice was initially driven by the enhanced SEM contrast achieved with neodymium.

In this study, we aimed to describe the structural, elemental chemical, and probable origin of zones of extreme lanthanoid (neodymium) accumulation that form during preparation for SEM analysis in corneal fibroblasts filopodia.

## 2. Results

### 2.1. Characteristics of Neodymium Staining of Filopodia in Cell Culture

We found that the extreme brightness and contrast zones in the filopodia of keratocytes are unique to neodymium treatment. The filopodia of untreated keratocytes do not exhibit zones of extreme brightness (Appendix A).

Lanthanoid-enhanced SEM revealed three distinct patterns of extreme neodymium staining in fibroblast filopodia (Figure 1). All three patterns represent asymmetric staining within a visually identifiable, thin, sharp, and extremely bright barrier zone, which is located perpendicular to the filopodial axis, as indicated by the white lines in Figure 1, denoting the barrier zone. The visually distinguishable barrier zone strongly limits the spread of the stain toward the cell body. The white arrows in Figure 1 illustrate the asymmetry of the staining pattern indicating asymmetrical increases in brightness along the length of the filopodia. These staining patterns were observed in various contexts of filopodial function and intercellular interaction.

In filopodia forming intercellular contacts, two primary staining patterns were identified. The first pattern (Figure 1a) is characterized by asymmetric staining regions with an extreme contrast zone within the filopodia, featuring a distinctive visually detectable barrier positioned perpendicular to the filopodial axis. This filopodial region presents a relatively flat morphology. The second pattern (Figure 1b), also observed in filopodia engaged in intercellular contact, displays asymmetric staining regions with a notable conical enlargement at the zone of maximum contrast intensity.

A third distinct pattern was observed in the distal regions of filopodia not engaged in intercellular interactions (Figure 1c). Unlike the first two patterns, this one is specifically associated with individual filopodia, where asymmetric staining is observed at a visually detectable barrier zone positioned perpendicular to the direction of single filopodial outgrowth. This pattern suggests a potential role in environmental sensing or preparation for future cellular interactions.

Based on these visual observations, as well as our previous findings [14,15,16,17] and the known chemical properties of lanthanoids described by other researchers [18,19,20], we hypothesize that neodymium staining may reflect variations in cell local phosphate anion concentrations in the presence of physical or physicochemical barriers corresponding to the observed visual barriers. To verify this hypothesis, we conducted additional chemical analytical measurements, which are detailed in the subsequent Results Section.

### 2.2. The Decoding of the (Bio)Chemical Nature of Extreme Contrast Zones in Filopodia—Data from Semi-Quantitative Chemical Microanalysis Using Scanning Electron Microscopy (SEM) Coupled with Energy-Dispersive X-ray Spectroscopy (EDS)

The brightness (measured in Gray Scale Units) of neodymium-stained cells observed in SEM-BSE images was found to be proportional to the local phosphorus content determined by energy-dispersive X-ray spectroscopy (EDS) across the entire cell body, and not limited to extreme contrast zones in the filopodia. A linear regression analysis showed a strong correlation, with a coefficient of determination (R^2^) of 0.77, allowing brightness to be used as a qualitative measure of phosphorus accumulation (Figure 2).

Chemical microanalysis of the extreme-contrast zones revealed a neodymium-to-phosphorus stoichiometric ratio approximating 1:1 (Figure 3e). This finding, coupled with the known chemical properties of lanthanoids [14,15,16,17,18,19,20] and the known presence of diverse phosphorus species in cellular environments [21,22,23], indicates the formation of simple neodymium phosphate (NdPO_4_) compounds and suggests a preferential accumulation of phosphate anions at specific barriers.

Notably, phosphate accumulation was observed at two distinct locations: within the contour of connected filopodia between adjacent cells, and in individual filopodia directly interacting with the extracellular matrix. This localized concentration of phosphate may be accompanied by a concomitant accumulation of sodium ions. The presence of sodium could potentially enhance phosphate availability to the cell by subtly modifying the solubility characteristics of the resulting phosphate compounds.

Interestingly, the levels of C, N, and O along the analysis line remained largely constant, suggesting that the observed phosphorus accumulation is a specific phenomenon rather than a general change in elemental composition.

Thus, the levels of C, N, O, P, Na, and Nd were evaluated. All other elements present in the sample were below the detection limit of EDS and could not be quantified. The semi-quantitative chemical analysis confirmed our earlier assumption that neodymium staining reflects variations in local phosphate anion concentrations within the cell. The extreme contrast zones were found to be areas of peak phosphate accumulation, which allows us to assume that these zones serve as localized depots of phosphate in the cells.

These findings provide novel insights into the spatial distribution of phosphate in filopodia and its potential role in intercellular communication and cell–matrix interactions. The observed patterns of phosphate accumulation may reflect specialized functional domains within filopodia, serving as sites of resource storage and regulation, possibly involved in signaling processes or structural modifications at cell–cell and cell–matrix interfaces.

### 2.3. Non-Colocalization of Mitochondria and Neodymium in Keratocyte Filopodia

The investigation into the spatial distribution of phosphate anions in filopodia labeled with neodymium revealed that the exact location of phosphate accumulation does not coincide with mitochondrial clusters. In the zone corresponding to the extreme contrast in the SEM-BSE image, only a weak nonspecific fluorescence is observed under light microscopy in the MitoTracker channel (Figure 4). This finding suggests that phosphate anion accumulation occurs independently of mitochondrial localization within the filopodia.

### 2.4. The Frequency of Filopodia Observations with Zones of Extreme Contrast in Culture with Induced Apoptosis

Further analysis demonstrated a higher prevalence of filopodia with barrier zones in which neodymium-labeled phosphate anions accumulate in cell cultures undergoing lipopolysaccharide (LPS) induced apoptosis (Appendix A). This phenomenon of specific neodymium-labeled phosphate anions accumulation was particularly pronounced between relatively healthy cells in LPS-treated culture and those exhibiting advanced stages of apoptosis (Figure 5). The increased number of such filopodia during apoptosis may indicate a role for phosphate anion accumulation in the apoptotic process or in the communication between dying and healthy cells.

To quantify this, we analyzed the ratio of filopodia with observable barrier zones to the total number of filopodia. The quantification was performed on BSE images of keratocytes, both under normal conditions and during induced apoptosis, using a direct and blinded method performed by trained students to reduce observer bias. In keratocytes under normal conditions, an average of 14% of filopodia were found to possess visible barrier zones, while this ratio increased to 24% under apoptotic conditions. Similar observations were made in other cell types and conditions, supporting the broader relevance of the phenomenon, as highlighted in the introduction. The calculated ratios for each cell type and condition are presented in Appendix A.

These findings suggest that the accumulation of phosphate at the intercellular barrier zones could be part of a mechanism by which healthy cells manage and redistribute essential resources from apoptotic cells. A more detailed discussion of this mechanism can be found in the Section 3.

## 3. Discussion

This presents an analysis of the properties of lanthanoids as cell stains, with a focus on zones of extreme staining accumulation within filopodia, using corneal fibroblasts as a model. Based on their chemical characteristics, lanthanoids in the cell can label both the phosphate residue and calcium [18,19,20,24,25,26]. In this study, we used a staining solution based on neodymium chloride, but previously we tested other lanthanoid elements that are chemically similar and exhibit similar distribution patterns, and whose behavior is consistent with the findings reported in the literature. 

It is important to note that neodymium is known to be a very effective substitute for calcium in a variety of systems due to its similar ionic radius. During our experiments, neodymium likely replaced calcium at its typical binding sites. This substitution occurs because neodymium has a higher binding affinity compared to calcium, allowing it to occupy calcium’s binding sites more effectively [27]. Consequently, by the time of visualization, the local concentration of calcium in the regions labeled by neodymium had dropped below the detection limit of the EDS technique (0.1 wt%). This explains why calcium was not detected in our spectra despite its known association with phosphate residues and other cellular components.

Given these circumstances, it is not possible to unequivocally determine the labeling of calcium with neodymium. Nevertheless, our experiments provide clear evidence for the labeling of phosphate residue with neodymium (Section 2.2, Figure 3).

The key question is whether we label with neodymium the free phosphate anion or phosphate residue that is bound to organic molecules? Since one of the dominant lobes in corneal cells is occupied by phosphate in the free form, it is most likely that we label it [21,22,23]. The laws of kinetics also support this: lanthanoids bind to the free residue more rapidly and react more slowly and less effectively with phosphate groups of organic molecules.

It should be noted that many phosphate-containing organic compounds, such as nucleic acids and ATP, are present in cellular environments and are potential candidates for neodymium binding. Another such compound mentioned in the literature is inositol triphosphate (IP3), which contains three phosphate residues and is a signaling molecule known for its role in the functioning of distal intercellular contacts. Although it is theoretically possible that we label phosphate-containing organic compounds, it is less likely compared to the direct labeling of free phosphate. This is because such binding would require additional time for the hydrolysis of phosphoester bonds, the release of phosphate residues from the organic molecule, and the subsequent formation of neodymium phosphate (NdPO_4_) [28,29,30]. It is also possible that the neodymium ions may form insoluble complexes directly with the organic molecule, which contains the phosphate anion [31,32]. However, based on our experimental data, the nearly equimolar ratio of phosphorus to neodymium (Figure 3e) is more consistent with the formation of simple neodymium phosphate, rather than complexation with larger organic molecules.

Special attention should be given to the causes of the formation of a distinct barrier zone where the phosphate anion accumulates within filopodia. When considering the interaction between two cells, it may be assumed that there is an extreme concentration at the physical barrier of the phospholipid membrane. However, this hypothesis cannot explain the existence of visible barrier zones within some single filopodia, which are used by cells for adhesion.

Another hypothesis suggests that the creation of this distinct barrier zone is due to the formation of a physicochemical barrier along the intracellular pH gradient. It is known that the distal regions of cell processes may be more acidic in relation to the cell body [33,34,35]. Based on this, it can be assumed that in single filopodia, precipitation of neodymium phosphate was observed along a pH gradient. Phosphate, which is more soluble in acidic environments (in this case, neodymium phosphate), loses solubility as it migrates toward the more alkaline cell body and precipitates at the solubility limit barrier.

According to the same principle, the availability of phosphate anions is regulated in untreated NdCl_3_ cells. In filopodia, phosphate residues are more mobile, but as they move toward the cell body, phosphate anions become less mobile and accumulate more and more. When a certain pH level is reached (presumably > 7.2), phosphate anions restore the ability to bind to organic molecules for participation in phosphate biogenic transport, and colloids begin to form with cations. That is, it becomes immobilized and accumulates at the physicochemical barrier, which creates an extreme BSE-contrast zone in this area due to the weight of neodymium.

Indeed, under acidic conditions, many molecules undergo protonation, which reduces their reactivity and the efficiency of the addition of phosphate groups. Under more alkaline conditions, there is an increase in the availability of substrates for reactions, which generally leads to a more effective attachment of phosphate groups to organic molecules. Thus, the accumulation of phosphates in filopodia at the physicochemical barrier along the pH gradient results in the formation of a phosphate depot, which contributes to a further regulated cellular transport mechanism, allowing the cell to control and utilize these phosphates as a supply for metabolic and signaling processes.

At the same time, the accumulation of phosphorus is probably accompanied by sodium accumulation (Figure 3d,e). Regarding the presence and accumulation of sodium, it is important to approach this finding with caution due to the nature of the EDS detection method. Sodium is a relatively light element, and therefore its detection in such an environment can involve some uncertainties and may be associated with measurement errors.

As for the possible mechanisms of sodium accumulation, it should be noted that, due to its chemical properties, sodium ions do not accumulate simply due to a pH gradient. Sodium ions tend to remain in a highly mobile state under varying pH conditions, which limits their ability to accumulate directly due to pH changes alone. However, they may accumulate with already immobilized phosphate anions. These interactions contribute to phosphate availability for cellular processes, as sodium slightly reduces the solubility of inorganic phosphates. This is consistent with the known role of sodium-stabilized colloids in the formation of crystalline calcium phosphate under conditions that shift from an acidic to an alkaline environment.

An alternative hypothesis explaining the simultaneous detection of phosphates and sodium in the observed zones could be based on the activity of specific sodium-phosphate cotransporters (such as Na/Pi cotransporters PiT-1 and PiT-2). These cotransporters are known to play a significant role in sensing extracellular phosphate and maintaining cellular phosphate balance [36,37,38]. The observed accumulation of sodium could result from a potentially high density of these cotransporters on the cell membrane. However, the ratio of cell membrane surface area to cell volume is relatively small, which suggests that even a very high density of cotransporters would not lead to the level of sodium and phosphorus accumulation that we observed. Additionally, the observed 1:1 Na/Pi stoichiometry suggests a physicochemical nature of accumulation, as the functioning of Na/Pi cotransporters typically involves other stoichiometry [37].

Thus, a depot of inorganic phosphate, potentially stabilized by sodium ions, forms within the filopodia at the physicochemical barrier. Moving toward the cell body in the alkaline zone of the main cellular volume, instead of spontaneous migration of phosphorus in the anionic form, regular transport begins to predominate. In this instance, the barrier zone restricts the free flow of phosphate anions from the distal part of the filopodia, which may be regarded as a regulatory mechanism (Figure 6).

Based on the aforementioned concept of phosphate deposition in a concentration gradient, it is possible to consider the specific role of extreme accumulation zones that form in filopodia between cells. It has been established that in cells whose death is carried out through regular apoptosis, the cytoplasm becomes enriched with phosphate ions [39]. The high acidity of the cytoplasmic environment during apoptosis facilitates the maintenance of high concentrations of inorganic phosphate (Pi) in an available “convenient” form for transport [40]. It is possible that, when the more “acidic” filopodia of an apoptotic cell interacts with the more “alkaline” filopodia of a normal cell, the latter may accumulate a phosphate residue due to the pH gradient. As a result, the energy substrate (Pi) may be “pulled” from the dying cell and transferred to the healthy one. Furthermore, even if a physical barrier exists in the form of a phospholipid membrane at the junction between cells, phosphate accumulation would occur not in the membrane zone, but in the zone of reduced phosphate solubility on the pH gradient. This process is schematically represented in Figure 6.

Since cells interact at the site of focal adhesions, it would be important to have some information on the relationship between the stained structures and focal adhesion. In our study, we did not use immunocytochemical analysis for the detection of focal adhesion proteins (such as vinculin), so we do not have direct evidence of how focal adhesions relate to the structures we observed. However, since focal adhesions are well-known sites of cell–matrix interactions, it is theoretically possible to consider a potential relationship between the observed neodymium accumulation zones and focal adhesions.

Focal adhesions mainly consist of proteins that do not suggest a high level of binding with neodymium ions. However, it is also known that phosphoinositides containing phosphate groups play a key role in the functioning of focal adhesions [2,3,41,42,43]. Nonetheless, as we previously mentioned in the discussion, it is kinetically unlikely that neodymium would actively bind to phosphate groups in organic molecules, including phosphoinositides. Moreover, the locations of extreme neodymium accumulation were often found in filopodia close to the cell body, which differs from the typical location of focal adhesions at the tips of filopodia. Thus, the extreme accumulation of neodymium observed in our study is more likely related to Pi accumulation under pH gradient conditions, rather than specific molecular interactions with components of focal adhesions. Nevertheless, it is worth noting that some processes related to focal adhesion functioning (such as cytoskeleton assembly) may require large amounts of free phosphate residues. Therefore, it is theoretically possible and logical that the phosphate depo we observed may be functionally linked to focal adhesion activity, even if not directly caused by molecular interactions with their components.

Although numerous sites of cell–matrix and intercellular interactions exist, both on the cell body and filopodia, the heavily stained areas (the extreme neodymium accumulation zones) are limited. Not all filopodia are involved in phosphate transfer simultaneously, and this dynamic process depends on local physicochemical conditions, such as pH gradients along the filopodia, favoring neodymium binding during staining. Therefore, the limited number of extreme neodymium accumulation zones in filopodia is not unexpected, but rather reflects the specific and dynamic nature of cell–matrix and intercellular interactions.

One of the mechanisms of energy transfer between cells involves the direct or transmembrane transport of mitochondria from a cell that is under normal conditions to a recipient cell under stress. This process can help delay the apoptosis of the recipient cells [43]. In this case, how does the transfer of mitochondria relate to the proposed transfer of phosphate residue? Morphologically, the barrier zone at which phosphate anions accumulate in filopodia labeled with NdCl_3_ does not coincide spatially with clusters of mitochondria (Figure 4).

What can we conclude from this? Is it possible that a normal cell may pump out a phosphate anion from an apoptotic cell (including one released from destroyed mitochondria) and in return give mitochondria? This requires further investigation, but (if our interpretation of the results is correct), phosphate “cannibalism” at the Pi level, when a healthy cell within a population extracts phosphorus from a dying cell, has been observed directly (Figure 5b and Figure 7). It seems logical to us that those cells at the tissue level would not allow such a valuable resource to disappear unchecked.

Therefore, it is possible to consider a model where the transfer of a phosphate ion from an apoptotic cell to a normal one, and the converse transfer of mitochondria from normal to apoptotic cells, may be interconnected processes that help maintain metabolic balance under stress conditions. An apoptotic cell has lost its functions and energy, but it may have accumulated phosphates in a form that is accessible. Excess phosphorus is also released from dead mitochondria. The difference in pH between apoptotic and normal cells creates a gradient that facilitates the transfer of phosphate anions from apoptotic to normal cells.

A healthy cell that receives phosphate anions improves its energy status, which can contribute to its ability to function effectively. At the same time, the transfer of phosphate anions may serve as a signal to healthy cells that a neighboring cell is undergoing stress or apoptosis. In response to this signal, a healthy cell may activate mitochondrial transmission mechanisms, which could either aim to save the apoptotic cell or help the stressed cell to temporarily maintain its metabolic processes in order to recycle residual waste products and prevent further damage to surrounding tissues.

## 4. Materials and Methods

### 4.1. Cell Culture

A primary cell culture of human corneal stroma was prepared using the explant method from a corneoscleral ring obtained from a cadaver. The fragments of the paracentral region of the corneal stroma were placed on the surface of a culture plastic.

The cells were cultured in RPMI-1640 medium (Gibco, Grand Island, NY, USA) supplemented with GlutaMAX^®^, antibiotics—penicillin 100 U/mL, streptomycin 100 μg/mL (all from Gibco, USA), and 10% of fetal bovine serum (HyClone, Logan, UT, USA) under standard conditions (37 °C, 5% CO_2_, and 100% relative humidity). The culture medium was changed every 2–3 days, and when the cells reached confluence, they were passaged. Cells from the 3rd passage were used for the study. In some samples, apoptosis was induced by adding LPS (L4391 Sigma, San Diego, CA, USA) at a concentration of 1 μg/mL [44]. The concentration of LPS was selected in such a way that at least 10% of apoptotic cells were present in the population, so that they could be detected by visual inspection. Detection of apoptosis was performed on a flow cytometer BeamCyte 1026M (VDO Biotech, Suzhou, China) after staining for annexin V and propidium iodide (21172 Apoptosis Detection Kit, Lumiprobe RUS Ltd., Moscow, Russia).

### 4.2. Preparation, Visualization Using SEM, and Semi-Quantitative Chemical Microanalysis

Prior to imaging on a scanning electron microscope, samples on culture plastic were processed through solutions from the BioREE-A lanthanoid staining set (Glaucon LLC, Moscow, Russia) in accordance with the manufacturer’s instructions [45].

The images were acquired using a Zeiss EVO 10 scanning electron microscope (Zeiss, Oberkochen, Germany) that was equipped with a LaB6 cathode. A low-vacuum mode (70 Pascal) was employed at an accelerating voltage ranging from 20 kV to 25 kV. The backscattered electrons were detected, allowing for the identification of regions with a higher concentration of relatively heavy elements.

Semi-quantitative chemical microanalysis was carried out using a Zeiss EVO 10 microscope and an energy-dispersive X-ray spectrometer (EDS) from Zeiss. The relative concentrations of chemical elements in the sample whose concentrations exceeded the detection limit of the EDS (>0.1 wt%) were measured: C, N, O, P, Na, and Nd. All other elements present in the sample had concentrations below the detection limit and were therefore not quantified.

### 4.3. Determination of the Mitochondrial Localization in Cell Culture

The localization of mitochondria in a cell culture with induced apoptosis was investigated. To this end, a cell culture previously treated with LPS was stained using MitoTracker Orange CMTMRos (M7510 Invitrogen, Carlsbad, CA, USA) and imaging using a fluorescent microscope AxioVert A1 (Zeiss, Germany) in the epifluorescence mode (FilterSet 15). The background non-specific glow in the MitoTracker channel was not removed in order to simplify navigations through the images and ensure their accurate alignment.

### 4.4. Combining Images from Scanning Electron Microscopy and Fluorescence Microscopy

The images obtained from light microscopy were combined with an image of the same area acquired using a scanning electron microscope. Geometric alignment was accomplished using visual markers and specialized software. In the combined image, the red channel represents the intensity of MitoTracker fluorescence, while the green channel represents electron backscatter intensity (BSE) from the scanning electron microscope (SEM) images. The additive nature of channel overlap suggests that regions simultaneously bright in both BSE and the MitoTracker fluorescence will appear yellow in the combined image.

## 5. Conclusions

In this study, we have described a novel phenomenon of extreme phosphate accumulation in fibroblast filopodia using neodymium-facilitated SEM. Our findings indicate the existence in filopodia of specialized barrier zones that regulate phosphate accumulation and transfer. These zones may represent key domains for cell communication and cell–matrix interaction, both during normal physiological states and apoptosis.

While some of our hypotheses require further investigation for confirmation, our study already demonstrates a simple and reproducible technique for visualizing inorganic phosphate ion (Pi) accumulation in cell filopodia using neodymium staining coupled with scanning electron microscopy (SEM). This method offers high contrast and resolution, enabling rapid tracking of phosphate transfer zones, both under normal and apoptotic conditions. Furthermore, the ability to highlight zones of Pi accumulation and exchange may serve as a powerful tool in developing therapeutic strategies aimed at restoring tissue energy balance and preventing further cellular damage. In particular, this technique could play a critical role in diagnosing and treating diseases associated with cellular energy imbalances, such as heart failure and myopathies. Future research will further explore the broader applications of this method in diagnosing metabolic disorders and improving therapeutic targeting.

## Figures and Tables

**Figure 1 ijms-25-11076-f001:**
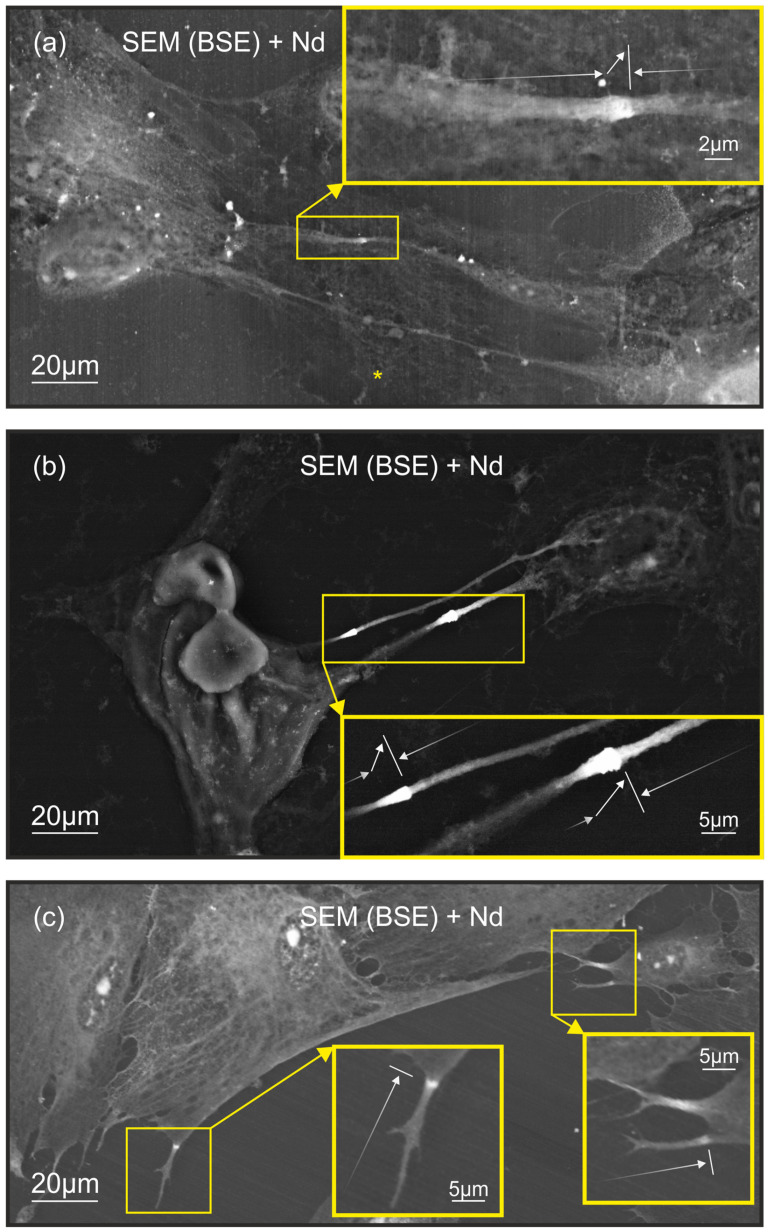
Keratocytes treated with NdCl_3_ solution on culture plastic: three types of asymmetrically dyed filopodia. The images are acquired using a scanning electron microscopy (SEM) equipped with a backscattered electron (BSE) detector. The yellow insets demonstrate enlarged images of keratocyte filopodia with observation of the extreme contrast barrier zone, which is denoted by a white line. The white arrows along the filopodia indicate a gradual increase in BSE brightness along the length of the filopodia. The white arrow, pointing at an angle toward the filopodia, indicates a rapid increase in brightness of the BSE: (**a**) Two connected keratocyte filopodia with a zone of extreme BSE contrast perpendicular to their axes. The area with the extreme contrast zone is not characterized by a significant variation in the diameter of filopodia. Asterisk in the same image indicates connected filopodia that do not exhibit an extreme contrast zone. (**b**) Connected filopodia of two keratocytes with an area with an extreme contrast zone, which is characterized by a cone-shaped thickening. (**c**) Single filopodia, without formation of intercellular junctions, with a distinct zone of extreme contrast perpendicular to them.

**Figure 2 ijms-25-11076-f002:**
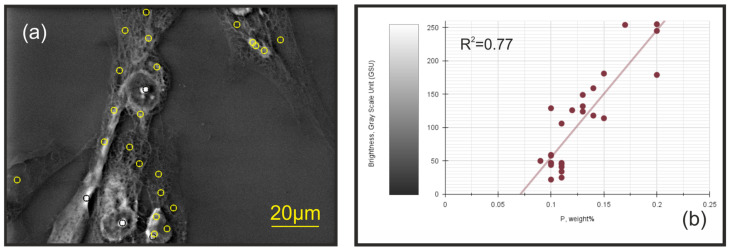
Correlation between phosphorus level and backscattered electron (BSE) brightness in different cellular regions: (**a**) Scanning electron microscopy (SEM) image (in BSE detection mode) of keratocyte filopodia treated with NdCl_3_ solution, where the different regions sampled for EDS analysis (including filopodia) are marked as distinct circles. The image was processed and exported using EDS software (Zeiss SmartEDX Standard. Version: 1.5.0012.0001). (**b**) A scatter plot where phosphorus content (in weight percent) is on the *x*-axis and SEM-BSE brightness (in Gray Scale Units) is on the *y*-axis. A color legend is included in the scatter plot to indicate Gray Scale Units (GSU), ranging from Low Brightness (0) to High Brightness (255).

**Figure 3 ijms-25-11076-f003:**
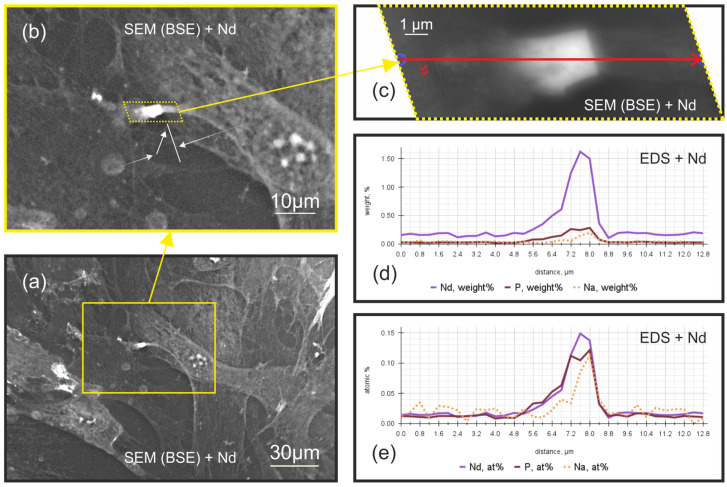
Scanning electron microscopy (SEM) images (in backscattered electron (BSE) detection mode) of keratocyte filopodia treated with NdCl_3_ solution and semi-quantitative chemical microanalysis data taken along a profile perpendicular to the extreme contrast zone (i.e., along the filopodia): (**a**) Keratocytes treated with NdCl_3_ solution placed on a culture plastic surface. An image was acquired using a SEM with a BSE. The yellow frame in the image indicates the area where filopodial junctions between two keratocytes are visible. (**b**) A pair of connected keratocyte filopodia, treated with an NdCl_3_ solution, are shown on a culture plastic. The yellow frame indicates the area where the junction of the keratocyte filopodial processes of the two cells forms a perpendicular linear region of extreme contrast, marked by a white, perpendicular line. The white arrows along the filopodia show a gradual increase in BSE brightness along their length. The white arrow, pointing at an angle toward the filopodia, indicates a rapid increase in the brightness of the BSE. (**c**) The filopodia junction of two keratocytes treated with an NdCl_3_ solution on a culture plate. The red arrow indicates the line along which a semi-quantitative chemical microanalysis was performed with a step of 0.4 μm, with a total of 33 analysis points. The BSE image was processed and exported using EDS software (Zeiss SmartEDX Standard. Version: 1.5.0012.0001). (**d**) The graph shows the change in the weight fractions of neodymium (Nd), phosphorus (P), and sodium (Na) along red arrow indicated in image (**c**). The *Y*-axis represents the weight fraction of each element in percent, while the *X*-axis indicates the distance from the beginning of the analysis in micrometers. The measurements were obtained using an energy-dispersive spectrometer (EDS). (**e**) The same in atomic fractions.

**Figure 4 ijms-25-11076-f004:**
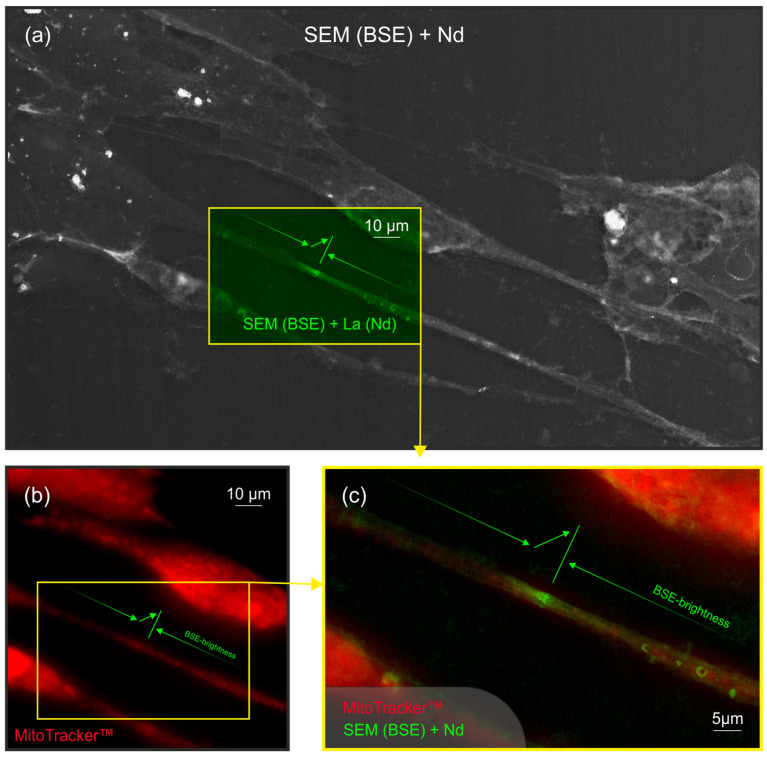
Localization of mitochondria and brightness of scanning electron microscopy (SEM) image (in backscattered electron (BSE) detection mode) of keratocytes filopodia. The fibroblast culture on plastic was visualized using two methods in sequence. The red channel represents the distribution of fluorescence intensity of the mitochondria as visualized by light microscopy. The green channel corresponds to the brightness distribution in the BSE image obtained from neodymium-enhanced SEM. In all images: the green line and arrows represent a change in brightness based on SEM (BSE) data: the line indicates the extreme contrast zone, the arrows along filopodia represent a gradual increase in BSE brightness along their length and the arrow at an angle to filopodia denotes a rapid increase in BSE to the extreme contrast zone: (**a**) The image was obtained using a SEM with BSE detection after neodymium staining. The yellow rectangle highlights the junction area between the filopodia of two keratocytes. A green channel has been highlighted for further combining. (**b**) The same picture. Keratocytes were treated with MitoTracker (MitoTracker Orange CMTMRos, M7510 Invitrogen, Carlsbad, CA, USA). The yellow rectangle in the figure corresponds to a similar area of the junction of filopodia from two keratocytes (as shown in image (**a**)). The red channel has been highlighted for further combining. (**c**) The resulting picture.

**Figure 5 ijms-25-11076-f005:**
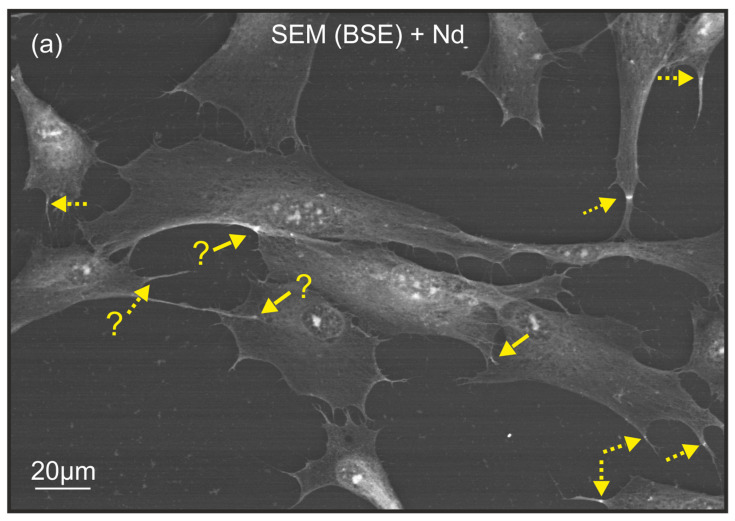
Keratocytes treated with NdCl_3_ solution on a culture plastic surface measuring 260 × 180 square meters. The images were obtained using a scanning electron microscopy (SEM) equipped with a backscattered electron (BSE) detector. The yellow continuous arrows indicate the connected filopodia of two keratocytes, with a clearly visible extreme contrast zone. Yellow dotted arrows indicate single filopodia with a clear zone of extreme contrast perpendicular to them. The question mark is used in cases where it is not possible to speak with certainty about intercellular communication through contact with filopodia: (**a**) Control cell culture: typical, uniformly bright, elongated keratocytes with normally shaped, round nuclei are observed. (**b**) Cell culture with lipopolysaccharide (LPS)-induced apoptosis: the keratocyte membrane exhibits characteristic apoptotic vesicles (some of which are indicated by the red arrows). The red dotted line limits the group of cells that have the most significant apoptotic changes. Keratocytes in this group show a relatively low level of brightness on the SEM-BSE image with neodymium staining. They have become rounded and their nuclei have lost their shape. The number of filopodia with extreme contrast zones, which are associated with an asymmetric increase in brightness, is higher in this group compared to the control group (**a**), particularly at the boundary of the zone with pronounced apoptosis.

**Figure 6 ijms-25-11076-f006:**
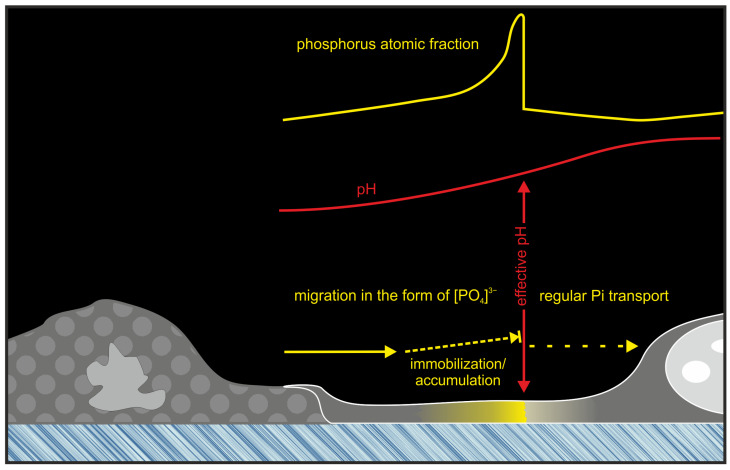
A scheme explaining the accumulation of free phosphate anions on the physicochemical barrier in the distal region of the filopodia.

**Figure 7 ijms-25-11076-f007:**
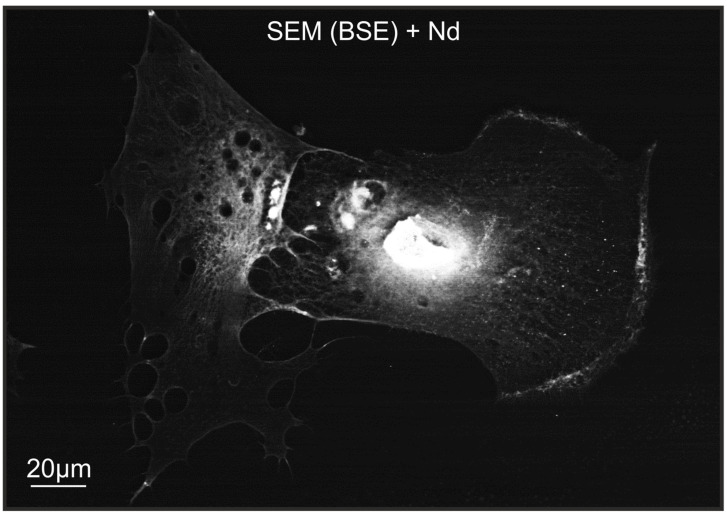
Phosphate “cannibalism” in close-up view. The keratocyte on the left is seen pulling Pi through filopodia from a keratocyte on the right that has previously undergone apoptosis. The scanning electron microscopy (SEM) image (in backscattered electron (BSE) detection mode) of keratocytes treated with NdCl_3_ solution on culture plastic.

## Data Availability

The data are not publicly available due to restrictions on their containing information that could compromise the privacy of research participants. Requests to access the additional data should be addressed to the following email: kravchik.mv@gmail.com.

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
