# Peer review of "Neodymium-Facilitated Visualization of Extreme Phosphate Accumulation in Fibroblast Filopodia: Implications for Intercellular and Cell–Matrix Interactions"

_ijms, 2024, doi:10.3390/ijms252011076_

Round 1
Reviewer 1 Report
Comments and Suggestions for Authors
The manuscript by Kravchik and coworkers describes the use of supravital lanthanoid staining for the visualization by Scanning Electron Microscopy (SEM) of intercellular and cell-matrix interactions and of phosphate accumulation in cellular filopodia. Data are also related to normal conditions and during apoptosis.
Lanthanoid staining are well known for their ability to associate to cell membranes allowing visualization of the surface structure, subsurface layer, and the inner structure of the biological sample.
1) Authors suggest that areas of intense staining may play a role in intercellular communication and cell-matrix interactions. Since cells interacts at the site of focal adhesions, it would be important to have some information on the relationship of the stained structures and focal adhesion. Authors should better explain this point
2) Moreover, since site of interactions are quite numerous, Authors should explain why there are only few heavily stained areas.
3) Line 117: “The brightness pattern of areas with extreme contrast zone in filopodia observed in SEM-BSE images were found to be proportional to the local phosphorus content determined by EDS, allowing brightness to be used as a qualitative measure of phosphorus accumulation”. How Authors measure the correlation between brightness and phosphorus concentration. How many measures were performed? Please clarify.
4) Authors should comment more extensively the presence/accumulation of sodium.
5) Apoptosis has been induced by LPS. How was demonstrated the occurrence of apoptosis, and how long were the cells exposed to LPS? Please add. The use of specific apoptotic marker is also advisable.
6) Line 212: Authors state that lanthanoids can label phosphate and calcium as well. Authors did not show any presence of calcium. Calcium can be clearly shown by EDS. Show some spectra obtained analyzing stained areas.
7) Line 226: It is possible that staining is labelling inositol 3P? The answer of Authors is yes. This is definitely an overstatement. There is no demonstration for the presence o I3P in these areas. Moreover, in cells there are many other components rich in phosphorus (nuclei acids for instance, ATP…..).
8) More data should be provided to demonstrate the presence of the barrier zone.
9) Line 309: “healthy cell within a population extracts phosphorus from a dying cell, has been observed directly (Figure 4, 5).” Figure 5 is a drawing, therefore it is not a demonstration. Figure 4, showing keratocytes, does not demonstrate extraction of Pi
Comments on the Quality of English LanguageSome sentences are not very fluent. Check also for typos.
Author Response
Dear Reviewer,
We sincerely thank you for your thorough review of our manuscript. Your insightful comments and suggestions have been invaluable in improving the quality and clarity of our work. We have carefully considered each point raised and have made significant revisions to address your concerns. Please find below our point-by-point response to your comments:
1) Authors suggest that areas of intense staining may play a role in intercellular communication and cell-matrix interactions. Since cells interacts at the site of focal adhesions, it would be important to have some information on the relationship of the stained structures and focal adhesion. Authors should better explain this point
In our study, we did not use immunocytochemical analysis for detection of focal adhesion proteins (such as vinculin), so we do not have direct evidence of how focal adhesions relate to the structures we observed. However, since focal adhesions are well-known sites of cell-matrix interactions, it is theoretically possible to consider a potential relationship between the observed neodymium accumulation zones and focal adhesions.
Focal adhesions mainly consist of proteins that do not suggest a high level of binding with neodymium ions. However, it is also known that phosphoinositides containing phosphate groups play a key role in the functioning of focal adhesions [PMID 26728462,38521786,34041035,31843255,30639111]. Nonetheless, as we previously mentioned in the discussion, it is kinetically unlikely that neodymium would actively bind to phosphate groups in organic molecules, including phosphoinositides. Moreover, the locations of extreme neodymium accumulation were often found in filopodia close to the cell body, which differs from the typical location of focal adhesions at the tips of filopodia. Thus, the extreme accumulation of neodymium observed in our study is more likely related to Pi accumulation under pH gradient conditions, rather than specific molecular interactions with components of focal adhesions.
Nevertheless, it is worth noting that some processes related to focal adhesion functioning (such as cytoskeleton assembly) may require large amounts of free phosphate residues. Therefore, it is theoretically possible and logical that the phosphate depo we observed may be functionally linked to focal adhesion activity, even if not directly caused by molecular interactions with their components.
In accordance with our response to this question, we have expanded our 'Discussion' section. (Lines 427-447)
2) Moreover, since site of interactions are quite numerous, Authors should explain why there are only few heavily stained areas.
Although numerous sites of cell-matrix and intercellular interactions exist, both on the cell body and filopodia, the heavily stained areas (the extreme neodymium accumulation zones) are limited. Not all filopodia are involved in phosphate transfer simultaneously, and this dynamic process depends on local physicochemical conditions, such as pH gradients along the filopodia, favoring neodymium binding during staining. Therefore, the limited number of extreme neodymium accumulation zones in filopodia is not unexpected, but rather reflects the specific and dynamic nature of cell-matrix and intercellular interactions.
In accordance with our response to this question, we have expanded our 'Discussion' section. (Lines 448-455)
3) Line 117: “The brightness pattern of areas with extreme contrast zone in filopodia observed in SEM-BSE images were found to be proportional to the local phosphorus content determined by EDS, allowing brightness to be used as a qualitative measure of phosphorus accumulation”. How Authors measure the correlation between brightness and phosphorus concentration. How many measures were performed? Please clarify.
We conducted semi-quantitative chemical analyses using EDS across multiple regions of interest, including cell areas within and outside of filopodia. Specifically, we performed over 20 separate measurements to determine the correlation between brightness and local phosphorus concentration. This extensive analysis consistently showed that the brightness observed in SEM-BSE images is proportional to the local phosphorus content, regardless of the specific region within the cell. This finding supports the reliability of using brightness as a qualitative indicator of phosphorus distribution.
We have slightly revised the statement you mentioned to better reflect the general nature of our observations. We have also included data and Figure 2 that demonstrate the correlation between brightness and phosphorus accumulation, (Lines 156-170)
4) Authors should comment more extensively the presence/accumulation of sodium.
Thank you for your comment. Regarding the presence and accumulation of sodium, it is important to approach this finding with caution due to the nature of the EDS detection method. Sodium is a relatively light element, and therefore its detection in such an environment can involve some uncertainties and may be associated with measurement errors.
As for the possible mechanisms of sodium accumulation, it should be noted that, due to its chemical properties, sodium ions do not accumulate simply due to a pH gradient. Sodium ions tend to remain in a highly mobile state under varying pH conditions, which limits their ability to accumulate directly due to pH changes alone. However, they may accumulate with already immobilized phosphate anions. These interactions contribute to phosphate availability for cellular processes, as sodium slightly reduces the solubility of inorganic phosphates. This is consistent with the known role of sodium-stabilized colloids in the formation of crystalline calcium phosphate under conditions that shift from an acidic to an alkaline environment.
An alternative hypothesis explaining the simultaneous detection of phosphates and sodium in the observed zones could be based on the activity of specific sodium-phosphate cotransporters (such as Na/Pi cotransporters PiT-1 and PiT-2). These cotransporters are known to play a significant role in sensing extracellular phosphate and maintaining cellular phosphate balance [PMID: 29233890, 30109818, 31768576]. The observed accumulation of sodium could result from a potentially high density of these cotransporters on the cell membrane. However, the ratio of cell membrane surface area to cell volume is relatively small, which suggests that even a very high density of cotransporters would not lead to the level of sodium and phosphorus accumulation that we observed. Additionally, the observed 1:1 Na/Pi stoichiometry suggests a physicochemical nature of accumulation, as the functioning of Na/Pi cotransporters typically involves other stoichiometry [30109818]. (Lines 365-389)
5) Apoptosis has been induced by LPS. How was demonstrated the occurrence of apoptosis, and how long were the cells exposed to LPS? Please add. The use of specific apoptotic marker is also advisable.
Measurements using a flow cytometer were performed to confirm the induction of apoptosis. The cells were exposed to LPS for 48 hours. The generally accepted markers of apoptosis, Annexin V and propidium iodide, were used for identification. The results clearly showed the presence of an increased number of cells in a state of late apoptosis after LPS treatment.
We have placed dot plot diagrams illustrating this in Supplementary materials (Figure S2).
The tools and reagents are indicated in the M&M section.
6) Line 212: Authors state that lanthanoids can label phosphate and calcium as well. Authors did not show any presence of calcium. Calcium can be clearly shown by EDS. Show some spectra obtained analyzing stained areas.
The absence of detectable calcium in our samples is consistent with the known behavior of neodymium as a very effective substitute for calcium in many systems. During the staining process, neodymium ions replace calcium ions due to their similar chemical properties, which leads to the displacement of calcium from its binding sites. Consequently, at the moment of visualization, the remaining calcium concentration in these areas is expected to be below the detection limit of EDS (0.1 wt%), which explains why no calcium was detected in the spectra.
Additionally, we have included this information at the beginning of the Discussion section: (Lines 296-307).
It is important to note that neodymium is known to be a very effective substitute for calcium in a variety of systems due to its similar ionic radius. During our experiments, neodymium likely replaced calcium at its typical binding sites. This substitution occurs because neodymium has a higher binding affinity compared to calcium, allowing it to occupy calcium’s binding sites more effectively. Consequently, by the time of visualization, the local concentration of calcium in the regions labeled by neodymium had dropped below the detection limit of the EDS technique (0.1 wt%). This explains why calcium was not detected in our spectra despite its known association with phosphate residues and other cellular components.
7) Line 226: It is possible that staining is labelling inositol 3P? The answer of Authors is yes. This is definitely an overstatement. There is no demonstration for the presence o I3P in these areas. Moreover, in cells there are many other components rich in phosphorus (nuclei acids for instance, ATP…..).
Thank you for your advice. Indeed, our statement regarding the labeling of the I3P molecule with neodymium may have seemed more definitive than intended. As demonstrated in the studies by Lock et al., 2016, IP3 plays a significant role in cellular communication, especially in the filopodia. In our discussion, we hypothesized that IP3 could potentially be labeled by neodymium in filopodia, given its important role in these cellular structures. However, we agree that, given the complexity of cellular environments and the difficulty of precise quantitative determination of phosphate-containing molecules, other phosphorus-containing compounds, such as ATP and nucleic acids, are also likely candidates for neodymium binding.
In light of your comment, we will revise our discussion to emphasize that other phosphorus-containing compounds, such as ATP and nucleic acids, are also candidates for neodymium binding.
We noted that (Lines 314-327):
It should be noted that many phosphate-containing organic compounds, such as nucleic acids and ATP, are present in cellular environments and are potential candidates for neodymium binding. Another such compound mentioned in the literature is inositol triphosphate (IP3), which contains three phosphate residues and is a signaling molecule known for its role in the functioning of distal intercellular contacts. Although it is theoretically possible that we label phosphate-containing organic compounds, it is less likely compared to the direct labeling of free phosphate. This is because such binding would require additional time for the hydrolysis of phosphoester bonds, the release of phosphate residues from the organic molecule, and the subsequent formation of neodymium phosphate (NdPO4) [27-29]. It is also possible that the neodymium ions may form insoluble complexes directly with the organic molecule, which contains the phosphate anion. However, based on our experimental data, the nearly equimolar ratio of phosphorus to neodymium (Figure 3e, former Figure 2e)* is more consistent with the formation of simple neodymium phosphate, rather than complexation with larger organic molecules
*Note: Due to the addition of a new figure (now labeled as Figure 2), all subsequent figure numbers have been adjusted accordingly (former Figure 2 is now Figure 3).
8) More data should be provided to demonstrate the presence of the barrier zone.
As far as we understand, you are requesting either additional evidence regarding the nature of the barrier or more quantitative confirmation.
In our study, the identification of the barrier zone was initially based on the observed staining patterns within the filopodia. We have clarified in our results that the barrier zone is initially and primarily identifiable through visual observations. (Lines 106-127)
Since direct detection of a physicochemical barrier is not possible, we rely on multiple measured characteristics of a visually detectable barrier zone, which we interpret as a possible physicochemical barrier zone where phosphate anions accumulate. This interpretation is supported by several facts mentioned in our study such as the peak increase in neodymium phosphate concentrations along the filopodia, and the association between these zones and areas of pH gradients, as described in the literature.
However, we appreciate your point regarding the need for more data. To strengthen our findings, we include in Section 2.4 and add to the Supplementary more detailed data showing the variability and frequency of these barrier zones across multiple samples. (Lines 258-266)
9) Line 309: “healthy cell within a population extracts phosphorus from a dying cell, has been observed directly (Figure 4, 5).” Figure 5 is a drawing, therefore it is not a demonstration. Figure 4, showing keratocytes, does not demonstrate extraction of Pi.
Thank you for pointing out the incorrect reference to Figure 5 (now labeled as Figure 6)* as a demonstration. We acknowledge that Figure 5 (now labeled as Figure 6)* is indeed a drawing, and not direct experimental data. We apologize for this oversight.
The statement that “healthy cell within a population extracts phosphorus from a dying cell, has been observed directly" was intended to refer to Figures 4b (now labeled as Figure 5b)* and 6 (now labeled as Figure 7)*, where we observe the interactions between apoptotic and healthy cells. The presence of high-contrast barrier zones with extreme brightness in the BSE-SEM images, which corresponds to elevated phosphorus content, supports the possibility of phosphate accumulation and transfer between cells.
For clarity, we have added the phrase 'if our interpretation of the results is correct' to the paragraph. (Line 473)
To support our interpretation, we now provide additional data (Figure2), demonstrating the correlation between BSE brightness and phosphorus content. These additional results offer stronger evidence in support of the thesis about direct observation.
*Note: Due to the addition of a new figure (now labeled as Figure 2), all subsequent figure numbers have been adjusted accordingly (e.g., former Figure 2 is now Figure 3, and so on).
We sincerely appreciate the time and effort you have invested in reviewing our work. Your feedback has been instrumental in elevating the quality of our research. We hope that you find our revisions satisfactory and look forward to your further comments.
Thank you for your consideration.

Reviewer 2 Report
Comments and Suggestions for Authors
Review of “Neodymium-Facilitated Visualization of Extreme Phosphate Accumulation in Fibroblast Filopodia: Implications for Intercellular and Cell-Matrix Interactions” by Kravchik et al. This interesting paper explores the use of supravital lanthanoid (neodymium) staining coupled with scanning electron microscopy (SEM) to visualize intercellular and cell-matrix interactions in corneal fibroblasts (keratocytes). The study identifies 3 distinct morphological patterns of neodymium staining in filopodia, revealing an asymmetric distribution of neodymium within a bright, sharp barrier zone. Semi-quantitative chemical analysis suggested variations in phosphate anion concentrations, with extreme phosphate accumulation in specific regions of filopodia. These patterns become more pronounced during apoptosis. The paper enhances understanding of phosphate transfer mechanisms in cells. I was impressed by the quality of this paper. It is very well written, and the authors findings are very interesting. I have several suggestions to improve this manuscript before publication:
Major comments:
11) One of the main findings of significance is that “Lanthanoid-enhanced SEM revealed three distinct patterns of extreme neodymium staining in fibroblast filopodia”. These results are shown in Figure 1. However, images of untreated keratinocytes (control cells) are not provided. It is unclear whether the presence of such extreme contrast zones is unique to Nd treatment.
22) Lines 83-92: it is unclear what the authors are referring to. Please supply references to specific figure panels. Where can we see these “patterns”? The authors are recommended to make their work easily understandable by the reader/reviewer.
33) While the introduction is generally well-written, the authors should bear in mind that IJMS is not a specialized journal focusing on microscopy. Some important aspects should be explained for a broader audience. Specifically,
a. More background information about “lanthanoid staining coupled with SEM” should be provided in general terms to introduce this topic to the reader.
b. How exactly does lanthanoid SEM “reveal the kinetics of biochemical processes occurring within the cell”? Please explain and support your statement with references.
c. How exactly does lanthanoid SEM “provides a snapshot of cellular metabolism”? Please explain and support your statement with references.
d. Why was neodymium a lanthanoid of choice in this paper? How does it perform compared to other lanthanoids?
e. What is the overarching problem (i.e. scientific question) the authors are trying to address in their research?
44) In Materials and Methods, catalog numbers must be provided for all reagents to improve reproducibility of the results.
55) In Lines 93-95, the authors suggest that Nd staining accumulates near (?) phosphate anions. Please explain to the reader how you made this assumption (e.g. cite a specific reference showing that Nd binds to/labels phosphate)? Otherwise, the author’s logic is unclear. The authors have tried to provide some sort of explanation in lines 121-132 (why so late in the text?), but these statements are not supported by experimental evidence or literature sources. Line 121 needs reference to a figure panel.
66) The authors claim that there is a “a higher prevalence of filopodia with barrier zones in which neodymium-labeled phosphate anions accumulate”. However, there is no quantification provided.
77) The authors claim that “Based on the findings of this study, it is not possible to unequivocally determine the labeling of calcium with neodymium”. The authors have not tried to address this question, as Figures 2d,e do not show Calcium analysis. Please delete or provide more experiments.
88) Figure 6 is not referenced in the text. A theory suggested in the discussion (313-320) is very interesting; it is frustrating that the authors have not attempted to perform more functional studies to test it. If real, this article would have had much more impact and provide a conceptual advancement. Currently, the authors’ preliminary results are of observational nature.
99) The conclusion section is rather vague. Please be more specific.
110) I noticed that some references are quite old (e.g. 1985, 1990, etc.). It is recommended to replace these obsolete references with more recent ones, if possible.
Minor comments
1) Line 50 – delete repeating word “resolution”
2) Line 52 – “the ultrastructure of cellular structures” => “cellular ultrastructure”
3) Line 59 – “it also presents challenges in terms of complexity and the need for specialized sample preparation methods” reference needed
4) Line 68 – “filopodia like” => “filopodia-like”
5) Line 192 – please delete “4)”
5) Please make sure all figures and panels are references in the correct places in the manuscript text. This will greatly improve the quality of your paper.
Author Response
Thank you for your thorough review of our manuscript. We greatly appreciate the time and effort you've invested in providing such detailed and insightful feedback. Your comments have been invaluable in helping us improve our work. We have carefully considered all of your suggestions and have addressed them in our revised manuscript. Below, you'll find our responses to your major and minor comments:
Major comments:
11) One of the main findings of significance is that “Lanthanoid-enhanced SEM revealed three distinct patterns of extreme neodymium staining in fibroblast filopodia”. These results are shown in Figure 1. However, images of untreated keratinocytes (control cells) are not provided. It is unclear whether the presence of such extreme contrast zones is unique to Nd treatment.
We have added wide-field images of untreated keratocytes to the Supplementary Materials to clarify whether the extreme brightness and contrast zones are unique to neodymium treatment. (Lines 103-105) Figure S1. Wide-field SEM-BSE image of keratocytes on culture plastic. The filopodia do not exhibit zones of extreme brightness.
22) Lines 83-92: it is unclear what the authors are referring to. Please supply references to specific figure panels. Where can we see these “patterns”? The authors are recommended to make their work easily understandable by the reader/reviewer.
Thank you for your comments. We have updated the text to include specific references to figure panels, arrows and lines ensuring that the staining patterns we describe can be identified by readers (Lines 106-127)
33) While the introduction is generally well-written, the authors should bear in mind that IJMS is not a specialized journal focusing on microscopy. Some important aspects should be explained for a broader audience. Specifically,
A. More background information about “lanthanoid staining coupled with SEM” should be provided in general terms to introduce this topic to the reader.
Thank you. To address your concerns, we have expanded the introduction to provide a clearer explanation. We have added the following background details based on the methodology described in Novikov [15].
This approach, based on the rapid visualization method described in our previous works, involves treating the cells with physiologically buffered solution containing ions of relatively heavy lanthanoids. This method does not require traditional SEM fixation and complete dehydration of the samples, simplifying the preparation process and preserving cellular structures. The staining by relatively heavy lanthanoids also provides high-contrast SEM-images without the need for additional heavy metal sputtering, which is common in conventional SEM. A particularly important aspect of this method is that the relatively heavy lanthanoids bind to intracellular components, enabling their visualization by SEM using back-scattered electron (BSE) detector, which allows for the observation of not only surface structures but also subsurface components. (Lines 64-74)
B. How exactly does lanthanoid SEM “reveal the kinetics of biochemical processes occurring within the cell”? Please explain and support your statement with references.
C.How exactly does lanthanoid SEM “provides a snapshot of cellular metabolism”? Please explain and support your statement with references.
B+C answer
We have expanded the introduction by including additional information and references to Subbot et al. (2019) to support our statements regarding that lanthanoid SEM "reveals the kinetics of biochemical processes within the cell" and "provides a snapshot of cellular metabolism.
We include that:
The supravital staining of cells with lanthanoids such as neodymium induces a temporary ametabolic state. This state effectively halts intracellular processes, such as the movement of organelles and RNA transcription, without causing permanent damage to the cell. By capturing cells in this ametabolic state, the technique allows for the detailed observation of cellular metabolism and morphology at a specific point in time, effectively providing a "snapshot" of cellular activity. This process is described in Subbot et al. (2019), who demonstrated that lanthanoid staining preserves the cell in a reversible ametabolic condition. This effect makes it possible to study the cell in a static metabolic state, revealing the kinetics of cellular processes by effectively pausing them at a given time. (Lines 75-83)
D. Why was neodymium a lanthanoid of choice in this paper? How does it perform compared to other lanthanoids?
We have expanded the introduction by including this information.
We include that: Neodymium was selected as the lanthanoid of choice after our previous extensive empirical testing of various lanthanoids. Among the tested lanthanoids, neodymium provided the maximum brightness and contrast in backscattered electron (BSE) imaging. We did not specifically investigate the underlying reasons for this, as our choice was initially driven by the enhanced SEM contrast achieved with neodymium (Lines 90-94)
E. What is the overarching problem (i.e. scientific question) the authors are trying to address in their research?
Regarding the overarching problem we aimed to address, we would like to emphasize that our study is primarily phenomenological. During the development of a novel neodymium-based staining technique for scanning electron microscopy (SEM), we observed an unexpected phenomenon of extreme neodimium accumulation in fibroblast filopodia. We considered it important to describe this phenomenon in detail, as it may have significant implications for understanding cellular processes.
As stated in the "introduction", we aimed to describe the structural, elemental chemical, and probable origin of zones of extreme neodymium accumulation in filopodial structures in corneal fibroblasts (keratocytes), which form during preparation for SEM analysis. At this point, we cannot address a broader scientific question
However, if we look at the research from a broader perspective, our work potentially provides a simple and reproducible methodology for visualizing inorganic phosphate residues (Pi) transfer. This technique allows for the precise identification of Pi accumulation zones, which can serve as markers for evaluating cellular metabolic activity. The ability to monitor these processes in near real-time opens up new opportunities not only for observation but also for the regulation of phosphate-related cellular mechanisms.
Thus, we propose that our visualization method can be utilized by future researchers as a tool for probing Pi accumulation as a marker for cellular energy status, offering insights into both normal and pathological states. We have elaborated on this in the new conclusion section (Lines 547-563)
44) In Materials and Methods, catalog numbers must be provided for all reagents to improve reproducibility of the results. – Done
55) In Lines 93-95, the authors suggest that Nd staining accumulates near (?) phosphate anions. Please explain to the reader how you made this assumption (e.g. cite a specific reference showing that Nd binds to/labels phosphate)? Otherwise, the author’s logic is unclear. The authors have tried to provide some sort of explanation in lines 121-132 (why so late in the text?), but these statements are not supported by experimental evidence or literature sources. Line 121 needs reference to a figure panel.
Thank you for pointing out the need for \clarification, regarding neodymium staining and its relation to phosphate anions. The chemical properties of neodymium have been documented in our previous studies. Specifically, we refer to the work of Novikov et al. (2015, 2018), which demonstrated that lanthanoid ions, including neodymium, tend to bind to phosphate anions. We recognize that these references (originally cited as references 13 and 14 in our manuscript) might not have been entirely clear to the international audience, as they are not in English.
To ensure greater accessibility for all readers, we will add more accessible references in English, including our previous studies in English (15, 16). Furthermore, our interpretation that neodymium selectively binds with phosphate anions is based on the well-known chemical properties of lanthanides, as discussed in other studies [18-20], which describe the high affinity of lanthanides, including neodymium, for phosphates, particularly in aqueous environments, making them suitable for binding and visualizing phosphate groups in cells.
It is worth noting that in Section 2.1, we describe results that are primarily related to visually observable characteristics of the cells stained with neodymium. Based on these visual observations, as well as our previous findings [14-17] and the known chemical properties of lanthanoids described by other researchers [18-20], we hypothesize that neodymium staining may reflect variations in cell local phosphate anion concentrations at the presence of physical or physicochemical barriers corresponding to the observed visual barriers. To verify this hypothesis, we conducted additional chemical analytical measurements, which are detailed in the subsequent sections of the results.
Accordingly, we have revised (Lines 128-134)
66) The authors claim that there is a “a higher prevalence of filopodia with barrier zones in which neodymium-labeled phosphate anions accumulate”. However, there is no quantification provided.
The quantification was performed on keratocytes, both under normal conditions and during induced apoptosis. In addition, we expanded our quantifications to include other types of cells, as noted in the introduction, to support the broader relevance of the phenomenon observed. The analysis was conducted by trained students using a direct, blinded method, thereby minimizing observer bias. For each fields of view, we calculated the ratio of filopodia with observable barrier zones to the total number of filopodia. The average values obtained for keratocytes in both normal and apoptotic states are included in the main text, while a comprehensive table specifying all cell types, fields of view, and observed ratios is presented in the Supplementary Material (https://docs.google.com/spreadsheets/d/1gceow3RTzKXEMUulLfJcwfqSUscv9NSGfz1cNZmQAHc/edit?usp=sharing.
We will add the following to Section 2.4:
To quantify this, we analyzed the ratio of filopodia with observable barrier zones to the total number of filopodia. The quantification was performed on BSE images of keratocytes, both under normal conditions and during induced apoptosis, using a direct and blinded method performed by trained students to reduce observer bias. In keratocytes under normal conditions, an average of 14% of filopodia were found to possess visible barrier zones, while this ratio increased to 24% under apoptotic conditions. Similar observations were made in other cell types and conditions, supporting the broader relevance of the phenomenon, as highlighted in the introduction. The calculated ratios for each cell type and condition are presented in the Supplementary (Lines 258-270)
77) The authors claim that “Based on the findings of this study, it is not possible to unequivocally determine the labeling of calcium with neodymium”. The authors have not tried to address this question, as Figures 2d,e do not show Calcium analysis. Please delete or provide more experiments.
The absence of detectable calcium in our samples is consistent with the known behavior of neodymium as a very effective substitute for calcium in many systems. During the staining process, neodymium ions replace calcium ions due to their similar chemical properties, which leads to the displacement of calcium from its binding sites. Consequently, at the moment of visualization, the remaining calcium concentration in these areas is expected to be below the detection limit of EDS (0.1 wt%), which explains why no calcium was detected in the spectra.
In light of your comment, we rewrite our discussion. We noted that:
It is important to note that neodymium is known to be a very effective substitute for calcium in a variety of systems due to its similar ionic radius. During our experiments, neodymium likely replaced calcium at its typical binding sites. This substitution occurs because neodymium has a higher binding affinity compared to calcium, allowing it to occupy calcium’s binding sites more effectively. Consequently, by the time of visualization, the local concentration of calcium in the regions labeled by neodymium had dropped below the detection limit of the EDS technique (0.1 wt%). This explains why calcium was not detected in our spectra despite its known association with phosphate residues and other cellular components. (Lines 296-307)
88) Figure 6 is not referenced in the text. A theory suggested in the discussion (313-320) is very interesting; it is frustrating that the authors have not attempted to perform more functional studies to test it. If real, this article would have had much more impact and provide a conceptual advancement. Currently, the authors’ preliminary results are of observational nature.
Thank you for recognizing the interest in the theory suggested in the discussion (lines 313-320). Your positive feedback reinforces our belief in the significance of our findings. While we agree that additional studies could indeed strengthen the proposed theory and increase the impact of this work, we believe that the current observational results provide significant value.
It is important to note that our current research primarily aims at describing an observed phenomenon—one that, to the best of our knowledge, has not been reported previously. We believe that the observational nature of our current findings does not diminish their importance, but rather sets the stage for more studies in the future. Our observations can help guide future experiments aimed at exploring these phosphate accumulation zones in various cell types and physiological or pathological conditions. We hope that our current work will inspire and support further research efforts that could test the functional relevance of the described phenomenon in greater detail.
We have also added the correct reference to Figure 6 (now labeled as Figure 7).
99) The conclusion section is rather vague. Please be more specific.
In this study, we have described a novel phenomenon of extreme phosphate accumulation in fibroblast filopodia using neodymium-facilitated SEM. Our findings indicate the existence in filopodia of specialized barrier zones that regulate phosphate accumulation and transfer. These zones may represent key domains for cell communication and сell-matrix interaction, both during normal physiological states and apoptosis
While some of our hypotheses require further investigation for confirmation, our study already demonstrates a simple and reproducible technique for visualizing Pi accumulation in cell filopodia using neodymium staining coupled with scanning electron microscopy (SEM). This method offers high contrast and resolution, enabling rapid tracking of phosphate transfer zones, both under normal and apoptotic conditions. Furthermore, the ability to highlight zones of Pi accumulation and exchange may serve as a powerful tool in developing therapeutic strategies aimed at restoring tissue energy balance and preventing further cellular damage. In particular, this technique could play a critical role in diagnosing and treating diseases associated with cellular energy imbalances, such as heart failure and myopathies. Future research will further explore the broader applications of this method in diagnosing metabolic disorders and improving therapeutic targeting.)
110) I noticed that some references are quite old (e.g. 1985, 1990, etc.). It is recommended to replace these obsolete references with more recent ones, if possible.
Thank you very much for your comment regarding the dating of some of our references. We agree that more recent literature is often preferred. However, the references in question provide detailed and exhaustive data on intracellular phosphorus, measured via phosphorus-31 nuclear magnetic resonance spectroscopy. These foundational studies continue to be cited in current research, which speaks to their continuing relevance. While we searched for more modern equivalents, we unfortunately could not find recent studies that provide the same level of information.
Minor comments
1) Line 50 – delete repeating word “resolution”
Done
2) Line 52 – “the ultrastructure of cellular structures” => “cellular ultrastructure”
Done
3) Line 59 – “it also presents challenges in terms of complexity and the need for specialized sample preparation methods” reference needed
Done
4) Line 68 – “filopodia like” => “filopodia-like”
Done
5) Line 192 – please delete “4)”
Done
5) Please make sure all figures and panels are references in the correct places in the manuscript text. This will greatly improve the quality of your paper.
Done
We believe that these revisions have significantly improved the clarity and quality of our manuscript. We hope that our responses and the revised manuscript meet your expectations. Thank you once again for your valuable feedback.

Round 2
Reviewer 1 Report
Comments and Suggestions for Authors
The manuscript has been extensively revised taking into consideration reviewer's criticism and suggestions
Reviewer 2 Report
Comments and Suggestions for Authors
All the reviewer's comments have been fully addressed.